# Porosity Reducing Processing Stages of Additive Manufactured Molding (AMM) for Closed-Mold Composite Fabrication

**DOI:** 10.3390/ma13235328

**Published:** 2020-11-24

**Authors:** Marquese Pollard, Phong Tran, Tarik Dickens

**Affiliations:** 1High-Performance Materials Institute (HPMI), Tallahassee, FL 32310, USA; marquese1.pollard@famu.edu (M.P.); phong.tran@eng.famu.fsu.edu (P.T.); 2Department of Industrial and Manufacturing Engineering, Florida Agricultural and Mechanical University-Florida State University College of Engineering, Tallahassee, FL 32310, USA

**Keywords:** additive manufacturing, composite tooling, composite fabrication, porosity

## Abstract

This article aims to merge two evolving technologies, namely additive manufacturing and composite manufacturing, to achieve the production of high-quality and low-cost composite structures utilizing additive manufacturing molding technology. This work studied additive manufacturing processing parameters at various processing stages on final printed part performance, specifically how altering featured wall thickness and layer height combine to affect final porosity. Results showed that reducing the layer height yielded a 90% improvement in pristine porosity reduction. Optimal processing parameters were combined and utilized to design and print a closed additive manufacturing molding tool to demonstrate flexible composite manufacturing by fabricating a composite laminate. Non-destructive and destructive methods were used to analyze the composite structures. Compared to the well-established composite manufacturing processes of hand lay-up and vacuum-assisted resin transfer molding methods, additive manufacturing molding composites were shown to have comparable material strength properties.

## 1. Introduction

Carbon fiber, glass fiber, and natural fiber composite materials are subject to increased worldwide demand through many industries, resulting in a predicted 130-billion-dollar industry in 2024, up from about 90 billion in 2019 [1]. Due to the expense associated with traditional composite material processing, alternative processing methods have recently been explored to reduce the cost, namely the time of autoclave and out-of-autoclave processes [2,3]. With the development of materials compatible with the various types of additive manufacturing (AM) technology, the spectrum of usefulness associated with AM platforms has increased considerably [4,5,6,7,8,9,10]. AM has been deemed a disruptive technology due to its design flexibility, custom material formulation, integrability, and mass customization. The economic benefit, reduced lead time, tooling design flexibility, and autoclave compatibility have led to this technology being adopted into the composite manufacturing tooling/ mold industry.

Material extrusion or fused filament fabrication (FFF) is the most widely used form of AM by researchers and enthusiasts due to the simple mechanical construction and a wide variety of material selection. Advanced material formulations are now customized for a specific application by various combining methods. These advances in material formulations have resulted in soluble polymers that are autoclave compatible. This compatibility has opened the door to additional functional applications in the low volume composite manufacturing field via FFF [11,12,13,14,15]. Autoclavable processes provide state-of-the-art techniques for producing high-quality composite structures (e.g., <2% defect). However, expensive traditional materials (i.e., invar and aluminum [16]) used to create autoclave resistant molds incur long processing times associated with forming master molds for high volume production. AM tooling and molding considerations have provided insight into new ways to produce composite structures at a fraction of the cost compared to using existing subtractive computer numerical control (CNC) methods to create tooling and molds [17]. Researchers have recently demonstrated additive manufacturing molding (AMM) tooling capabilities that withstand the extreme autoclave environment for composite fabrication [13]. Providing high-quality autoclave composite structures using printable tooling at a reduced cost provides justification to continue improving composites structures processing technology due to the demand for composite materials. 

To fully utilize AMM, specifically for closed molding composite fabrication, the structure must possess the following characteristics: (1) thermally compatible printing material that closely mirrors the composite materials thermal expansion, (2) a rigid body/frame to support the processing load while maintaining desired geometry, and (3) a non-porous hermetic structure which enables a vacuum seal for the resin transferring molding (RTM) procedure. Researchers have proven the viability first two characteristics, and due to the desired closed molding application, we will focus on the third characteristic. The first two are related to the material’s intrinsic properties, while the porous nature can be altered based on material properties and AM processing stages and parameters. The porosity of printed parts can be controlled by introducing functional material [18,19,20,21] or altering processes parameters [22]. Altering processing parameters is less expensive and can be incorporated with an abundance of AM materials. AM processing parameters are split into three stages. The stages are pre-processing, intermediate processing, and post-processing. The pre-processing stages consist of geometric design, structure wall thickness, material conditioning, machine maintenance, simulation platforms, etc. These types of parameters are navigated prior to the initiation of the slicing program. Intermediate processing is next and mainly consists of slicer parameters such as printing speed, building orientation, layer height, infill density, infill pattern, chamber temperature, etc. Many researchers have studied factors such as these during the life of material extrusion technology [23,24,25]. Once the part is fabricated, post-processing is the final stage. Due to the variability in AM parts, it is beneficial to determine how post-processing methods can improve surface finishes and reduce part variability for the final application [26]. Post-processing can utilize several mechanical or chemical tools such as coating, sanding/grinding, and even computer numerical control (CNC) machinery [26,27].

Ultimately, each stage plays a fundamentally critical role in the components’ final performance for their intended application, either aesthetically, structurally, or functionally. Typically, the final application will dictate which stage is targeted during processing considerations. Here, we focus on one aspect of all three processing categories. Figure 1 displays the three processing categories with corresponding attributes. Once parameters are understood, the airtight AMM is then used to fabricate a composite laminate. Well established currently used composite manufacturing methods hand lay-up (HL) and vacuum-assisted resin transfer molding (VARTM), are both used to create additional composite laminates for comparison. HL is a composite fabrication technique that uses a roller or brush to distribute the resin matrix on the reinforcing material manually. Structural composite materials range from glass, carbon, kevlar, and natural fibers. One of the limitations of this method is the ending performance variation due to different fabricators technique. This holds especially when this method is used for in-situ repair of structures. A more consistent method is known as resin transfer molding (RTM). When a vacuum assists the transferring of resin into the enclosed area and compacts the fibers, this method is vacuum-assisted RTM. The second method, VARTM, first establishes an enclosure, usually with disposable plastic bagging material, to create a vacuum seal around the dry fibers. Once a vacuum is established, the resin is drawn into the enclosed area and slowly infiltrates the dry fibers until entirely saturated. This method has been investigated for in-situ repair by Ramos et al. [28] due to the consistency of performance compared to HL. The composites properties of the AMM, HL, and VARTM samples are evaluated utilizing non-destructive and destructive methods to understand the mechanical performance of each manufacturing type.

## 2. Materials and Methods

### 2.1. Vessel Design and Testing

#### 2.1.1. Additive Manufacturing of Pressure Vessels

All samples in this work were printed with a commercial desktop 3D printer (Lulzbot Taz 6) using a 0.5 mm nozzle. All printed samples were created using high impact polystyrene (HIPS) material (3DX Tech). The HIPS material was selected due to its solubility in the environmentally friendly solvent (limonene). The bed temperature and hot end temperature remained constant throughout the study per the manufacturer’s temperature recommendations of 100 °C and 235 °C, respectively. A concentric pattern of 100% infill and an extrusion multiplier of 1.0 was used to create the parts. The main parameters explored are wall thickness (WT) and layer height (LH), as they are believed to be significant design and processing parameters influencing part porosity. To evaluate the porosity of AM structures, a pressure vessel is designed to contain pressurized gas. Figure 2 displays CAD models of pressure vessels with different design parameters (WT) and processing parameters (LH). Table 1 provides a detailed breakdown of the parameters mentioned, such as the total processing time and the pressure vessel’s internal volume. Table 1 is primarily grouped by 0.15 mm and 0.40 mm LH. Despite the combination of layer heights and wall thickness, all vessels maintained a constant volume showed in Table 1. Due to the closed molding tooling application, a printed part’s ability to hold a vacuum or pressure is desired.

#### 2.1.2. Leak Testing Procedure

Two variations of tests (qualitative image analysis and quantitative water displacement) were performed on untreated samples to determine the effect of design and processing parameters on the printed component’s porosity. Figure 3A provides a visual of the setup that was used to collect the visual qualitative data. The vessel is connected to a portable air compressor (Campbell Hausfeld EX 1001, Harrison, OH, USA) and submerged underwater to visually detect the number of defects and their surface locations within the part after printing. A high-speed camera (Vision Research Company Phantom v 210, Wayne, MI, USA) with a Zeiss 50 mm f/2 Makro-planar lens was used to capture the escaped air due to leakages. The quantitative experimental setup used to determine the volumetric flow rate of the air leakage is visually described in Figure 3B. The vessel was inserted into the inverted graduated cylinder to collect leaked air at different input pressures. The total amount of air contained in 30 seconds was then measured on the graduated cylinder. Due to the inherent defects accumulated during the fabrication process, it is expected that the WT thickness can compensate for the lack of bonding within a single layer [23,29,30,31,32]. The LH was also investigated as a means of coping with the fabrication defects. The smaller LH was expected to outperform the larger due to material streams embodying a more elliptical shape. This elliptical shape leads to improved adhesion between lines and layers [33].

### 2.2. Mold Design and Testing

#### 2.2.1. Additive Manufacturing of Mold/Tooling

Molds were CAD designed and additively manufactured using HIPS. The optimized pre-processing and intermediate-processing parameters were used to design and build the structure. These parameters are attributed to the lowest leakage rate from the vessel pressure experiment. The mold was designed with an internal cavity where dry fibers and the remaining lay-up would be placed. An inlet and an outlet port for tube connection for resin transfer molding (RTM) process were also included in the design. The square mold’s footprint was 304.8 mm × 304.8 mm and had a height of 12.7 mm (12 × 12 × 0.5 in). Since we want to compare AMM to additional composite fabrication methods, the simple planar geometry was used to create a panel laminate structure for non-destructive and destructive evaluation.

#### 2.2.2. Mold Post-Processing

Due to the variability of AM processing, understanding the conditions of the post-processing techniques is necessary. Vapor bathing is a common chemical post-processing method used to heal the porous and rough surface of extrusion printed parts [34,35,36,37]. Here, we used a standard approach that can be incorporated regardless of the printed material. A brush-on exterior coating (sealing) method is used as the post-processing method to seal all remaining pores on the molding structure’s exterior to create a completely hermetic part. The material used to coat the mold is a thermoset polymer Armorstar VE IVEXC410, a Vinyl Ester/DCPD blend infusion resin manufactured by Polynt Group. Once coated, the mold is now conditioned for the final application of closed molding composite tooling using VARTM technology. To ensure the post-processed printed structure was entirely sealed, a vacuum was connected to the mold’s outlet port while the input was clamped. A vacuum bag was then placed over the mold and sealed with tacky tape on a glass substrate. Once the vacuum was activated, the plastic bag remained inflated around the mold as long as the structure was seal was true. On the contrary, if the seal were compromised, the vacuum would seep through the porous areas on the printed structure and cause the plastic bag to form around the mold. A comparison study was created from the post-processed mold and the pristine (no post-processing) mold to verify the air permeability properties before and after treatment.

### 2.3. Composite Design and Testing

#### 2.3.1. Manufacturing of Fiber-Reinforced Composites

Since closed mold composite fabrication is desired using AM technology, one question that remains is the structural quality of the composite structure produced using AMM technology. In this work, we use three fabrication techniques to establish a comparative study to test the quality of the composite: HL, VARTM, and AMM. All samples, respective to their manufacturing method, were cut from a two-ply panel with a 304.8 mm × 304.8 mm footprint. The panel consisted of carbon fiber IM6 plain weave (0/90) with vinyl ester/DCPD blend as the matrix material. Each fiber ply has a nominal thickness of 0.167 mm per ply. Since all panels were fabricated with two plies; essentially, we are evaluating porosity, compaction, and fiber volume fraction of the composite system.

#### 2.3.2. Non-Destructive and Destructive Evaluation

Besides cost, one of the differences when comparing fabrication techniques is mechanical properties and quality. Here, we seek an understanding of the composite structure and mechanical strength quality by utilizing non-destructive and destructive equipment. Composites samples were cut using a wet saw according to ASTM D 3039 [38]. A Dolphicam handheld ultrasound imaging device (DolphiTech) was used to inspect samples prior to destructive testing. Destructive mechanical testing using materials test system (MTS) 858 tabletop system (25 kN) was utilized following Table 1 of ASTM D 3039 to assess the structural integrity of the composite samples.

## 3. Results and Discussion

### 3.1. Simulation Results

Simplified 3D was utilized to analyze the initial G-code file prior to printing. The program visualized the nozzle’s travel path during fabrication and showed a horizontal “fabrication seam” visibly forming along the vessel’s body. The seams are visually observed during the simulation and physical samples, as shown in Figure 4. As the nozzle approaches full circular completion, the machine moves slightly in the positive y-direction to create the next circle with an incrementally larger diameter. Due to the machine’s resolution, there is a small area where the extruded material does not have the same thickness as other areas due to travel speed and resolution. This is shown in Figure 4C. Figure 4D,E show the actual seam on the printed samples. Due to the natural color of the HIPS printed material, blue dye was used to aid in the visual of the seam area in Figure 4E. Figure 4F displays the response of the seam under internal pressure. As the thickness of the walls increases, the vessel’s rigidity increases, which also prompts enhanced air holding performance at the expense of time and additional material cost. 

### 3.2. Pressure Vessel Results

The vessel experiment yielded both qualitative and quantitative data. Visual qualitative data was gathered with the aid of a high-speed camera. With the aid of phantom camera control (PCC) software, the camera displayed porous areas within the vessel by air pockets erupting from the printed vessel. The six parts in Figure 5 display qualitative results from the vessel experiment. Frame images visually show the six vessel design types that were tested using the explained methods. All images were captured during the lower bound of the charging experiment at 0.275 MPa (40 psi). All vessels printed with 0.15 mm LH (Figure 5A–C) showed significantly less porous areas throughout the entire structure compared to their counterparts printed at 0.40 mm LH (Figure 5D–F). This was observed due to improved interlayer adhesion. A reduced LH corresponds to a smaller z height movement, promoting greater adhesion between layers due to elliptical shape. Visually comparing samples printed at a LH of 0.15 mm, the 6 mm WT sample showed fewer porous regions. A similar phenomenon was observed with the samples printed at a LH of 0.40 mm. Vessels with a WT of 6 mm showed a reduction of pores, which are also demonstrated by the erupting air pockets’ intensity. The porous structure displayed a more turbulent flow rate. It can be assumed this is due to the void’s size paired with the charging pressure. Increasing charging pressure correlates with a more intense airflow rate within the structure. This remained true while increasing pressure at 0.137 MPa (20 psi) increments. The intensity of the escaped air also directly correlates to the size and number of pores/voids within the structure, allowing air to escape at a faster rate. Quantitative results are shown in Figure 6, where the top and bottom plots correspond to 0.40 and 0.15 mm LH, respectively. The data collected here corresponds to the qualitative study. All vessels that were fabricated at 0.40 mm LH showed a significant increase in the volumetric flow of escaped air. The larger LH (0.4 mm) showed a greater tendency to delaminate between layers, which significantly reduced the air holding capacity independent of wall thickness. The 6 mm and 3 mm created with a LH of 0.40 mm leaked at a rate of 53.75 mL/s and 72.25 mL/s, respectively, at 0.413 MPa (60 psi) charging. Compared to their counterparts, the 6 mm, 3 mm, and 1 mm vessels with a LH 0.15 mm leaked at a rate of 0.07 mL/s, 1.04 mL/s, and 0.80 mL/s at 0.413 MPa, respectively. Comparing the six sets of designs, the volumetric flow rate is reduced by greater than 90% for 1 mm, 3 mm, and 6 mm wall thickness with a 0.15 mm LH compared to the larger 0.40 mm LH. When charging to 0.413 MPa, the 1 mm WT with a LH of 0.40 mm could not contain the pressure and exploded before it reached the remaining interval pressures.

The fabrication seam was identified as the initial failing point with all printed designs. The seam is formed due to the nozzle movement during the changing of travel lanes during the material deposition process. It was consistently found that these seams are the weakest or most porous areas on the printed component. At the lowest testing pressure 0.137 MPa, the seam area always displayed discontinuities first. Thicker wall design was expected to create a more ridge structure and compensate for any defects formed during fabrication. This was confirmed with the quantitative leak rate analysis. Both 6 mm WT samples outperformed the other WT samples in their representative LH category. Although faster printing time is associated with a larger LH independent of WT, the amount of post-processing required to create an airtight structure may outweigh the longer initial fabrication time, as shown in Table 1. Since post-processing is still needed to produce a hermetic structure, it is critical to understand if the additional initial processing time a worthy exchange. Minimum defects will inherently result in less post-processing time and effort due to producing a component with fewer defects.

### 3.3. Mold Results

Higher resolution and reduced lead times are the factors that lead to creating high-quality molds with speedy implementation. We selected the optimal parameter of 0.15 mm LH paired with a 1 mm WT for printing the mold, due to the high-resolution demand in this field and reduced fabrication time, as seen in Table 1. The printed mold acts similarly to a plastic bag used in the VARTM method. Post-processing was then performed using a brush-on coating method to seal printed defects. To ensure post-processing is effective, a comparison between unprocessed and processed mold was performed. The plastic bag was able to provide an enclosure for both the printed molds. As expected, the mold that was post-processed using brush-on coating treatment could seal all remaining pores on the surface. Once the vacuum was activated, the bag remained inflated for 24 h to show mold durability and displayed no signs of leakages (Figure 7B). The bag covering the unprocessed mold deflated until firmly fitting around the mold itself (Figure 7A). Post-processing clearly shows a viable inexpensive solution to heal defective components. Closely considering dimensional tolerances, one could also print the larger LH to reduce printing time further since post-processing is still required for our closed mold application.

### 3.4. Composite Results

#### 3.4.1. Composite Processing

Laminates, manufactured with HL and VARTM, visually looked pleasing with no apparent defects (resin-rich or dry) locations. The AMM laminate showed multiple small voids on the bottom of the laminate (contacting glass substrate), indicating that resin could not fully penetrate both fiber plies in some locations. Nominal sample thickness was measured and was found to vary for all methods. While HL and VARTM showed relatively similar thickness, AMM had an alarmingly larger thickness considering the uniform two plie lay-up. Due to the dry locations and access resin, it can be assumed that, resulting in increased nominal laminate thickness, AMM samples will not perform as well as HL and VARTM. This is because of the fundamental fiber volume fraction ratio, which directly relates to the composite structure’s performance.

#### 3.4.2. Non-Destructive Results

A non-destructive evaluation was utilized to inspect the internal structural integrity of the composite samples. Figure 8 displays representative c-scan and b-scan images from the non-destructive Dolphicam ultrasound imaging evaluation. The top images (c-scan) represent a planar type view of surface scans, while the bottom images (b-scan) represent the cross-sectional view from the corresponding c-scan. The heat-map represents the location, size, and depth of possible defects present within the structure. The color gradient relates to the magnitude of the potential defects and non-uniform areas. As the color intensity increases, the sound waves are interrupted due to an internal discontinuity. Since we are evaluating composites, it is expected to observe voids present within the structure sample. Both scans for the AMM sample clearly shows a highly porous section. Since all samples were scanned in the manufacturing orientation, the cross-sectional b-scan identifies this area as a noticeable surface defect. In general, the size and count of pores observed increased from HL to VARTM to AMM, respectively. As discussed in the next section, the samples’ strength correlates to the quality and defective areas observed.

#### 3.4.3. Destructive Results

A strain rate of 2 mm/min and 5 MPa (725 psi) tensile clamp pressure were used as test settings on the MTS 858 tabletop system. Figure 9 displays ultimate strength (MPa) and maximum strain data observed during testing for all three fabrication types. Four specimens where tested for each fabrication type. Average ultimate strength measured on the HL, VARTM, and AMM specimens were observed at 440, 410, and 339 MPa, respectively. Average modulus of elasticity measured on the HL, VARTM, and AMM specimens were observed at 19.43, 17.72, and 15.48 GPa, respectively. We conclude HL samples outperformed both VARTM and AMM techniques due to the fiber plies’ compaction, which resulted in a high fiber volume ratio and higher strength. The fiber volume ratio was calculated by utilizing the mass and density of both constituents. The fibers were weighed before infusion, then the total laminate weight after infusion and curing was reduce by the fiber weight yielding the weight of the matrix. The average laminate thickness of each technique was measured at 0.59, 0.61, and 0.75 mm, respectively. The hierarchy of the samples’ thickness from least to greatest follows the same trend as the strength performance, indicating the reduction in strength is related to the thickness of the samples. The excess of voids and resin present within the structure (Figure 8), the less load the specimens could endure. Since all samples were created with two fiber plies, the added thickness for the AMM samples is attributed to excess resin constructing the entire sample. AMM samples displayed lowest ultimate stress due to excessive resin distribution. This resulted from the composite structure yielding premature matrix dependent effects due to its poor fiber volume ratio rather than fiber dependent measured in both HL and VARTM. According to ASTM D 3039, both HL and VARTM samples experienced three of the eight failure modes which include angled, splitting, and grip failure modes. HL and VARTM also showed similar failure areas and failure locations, which was located in the middle for the gauge area and the top for grip/tab area. However, the AMM samples only experienced lateral failures in the gauge area in the middle of the specimen. Since both HL and VARTM both displayed the same failure modes, there is no clear correlation which supports failure modes specific to each of the three fabrication types.

## 4. Conclusions

In this study, AMM technology is utilized as a closed molding tool for composite manufacturing (CM). The sequence of developing an AM part is separated into three primary processing stages: pre-processing, intermediate-processing, and post-processing. We investigated one component from each stage for its contribution to fabricating an airtight hermetic AM part for closed molding. It was found that LH has a significant impact on the porosity of the printed specimen, therefore controlling its ability to contain pressurized air. By reducing the LH 62% from 0.4 mm to 0.15 mm, we reduced the volumetric flow rate by greater than 90%. WT also showed to have an impact on the porosity of the printed part. The largest WT examined 6 mm, outperformed its opponents 3 mm and 1 mm WT in their respective LH category. Due to the demand for reduced time and materials, a 1-mm WT was paired with 0.15 LH to create the AMM tool with the least number of pores in a short manufacturing processing time. Post-processing proved to be necessary to completely seal the tools’ exterior surface for closed molding CM. Once post-processed, the AMM tool was able to hold a vacuum to fabricate a composite laminate. Compared to composite laminates manufactured by well-established CM techniques, HL and VARTM, AMM laminates displayed tensile properties within 22% of the average ultimate strength values and 0.3% of maximum strain until failure. Despite reduced tensile values due to excess resin and dry locations from improper wetting, AM was shown to be controllable, flexible, customizable, and achieved the desired goal for CM. AM has demonstrated the capability to produce molding/ tooling structures at low cost, with reduced lead time with high flexibility.

## Figures and Tables

**Figure 1 materials-13-05328-f001:**
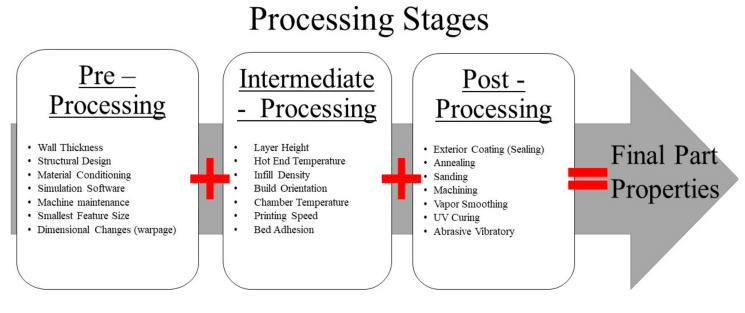
Additive Manufacturing (AM) three processing stages: pre, intermediate, and post processing.

**Figure 2 materials-13-05328-f002:**
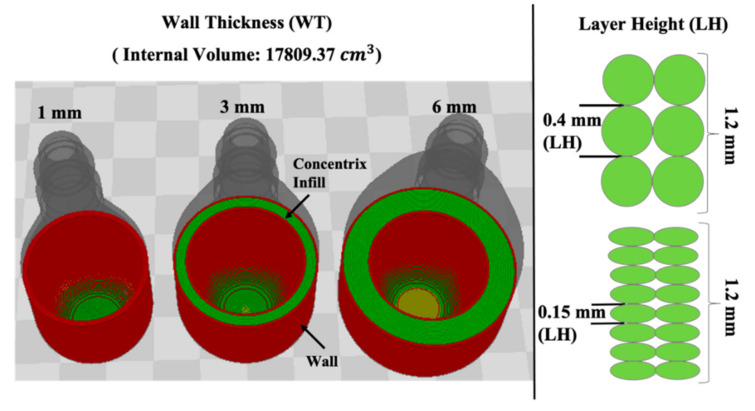
Render of cross-section of the vessel Wall Thickness and Layer Height design variables.

**Figure 3 materials-13-05328-f003:**
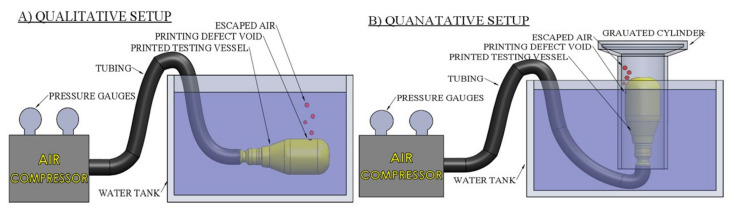
(**A**) Qualitative and (**B**) Quantitative vessel experimental setup.

**Figure 4 materials-13-05328-f004:**
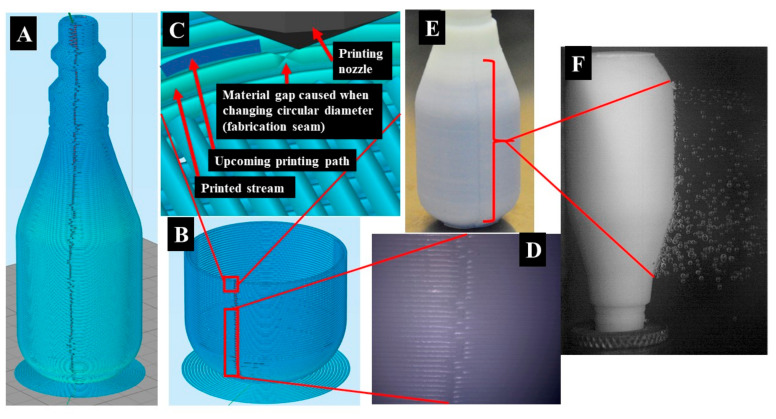
(**A**) Simplified 3D simulated vertical printing seam throughout entire vessel structure. (**B**) Simulated vertical and cross-section printed seam at layer 54. (**C**) Simulated zoomed cross-sectional layer seam nozzle, layer gap, upcoming path, and printed stream. (**D**) Physical zoomed vertical layer seam. (**E**) Physical vertical seam throughout entire structure. (**F**) Physical seam response under pressure at 0.275 MPa (40 psi).

**Figure 5 materials-13-05328-f005:**
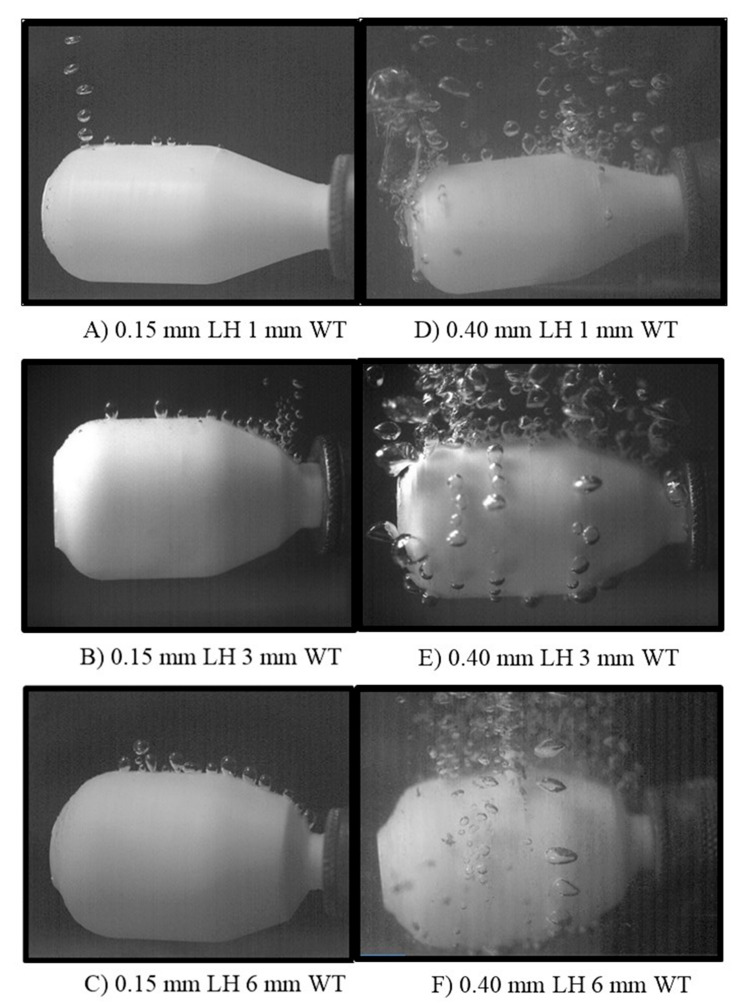
Qualitative vessel image analysis. (**A**–**C**) represent 0.15 mm LH. (**D**–**F**) represent 0.40 mm LH.

**Figure 6 materials-13-05328-f006:**
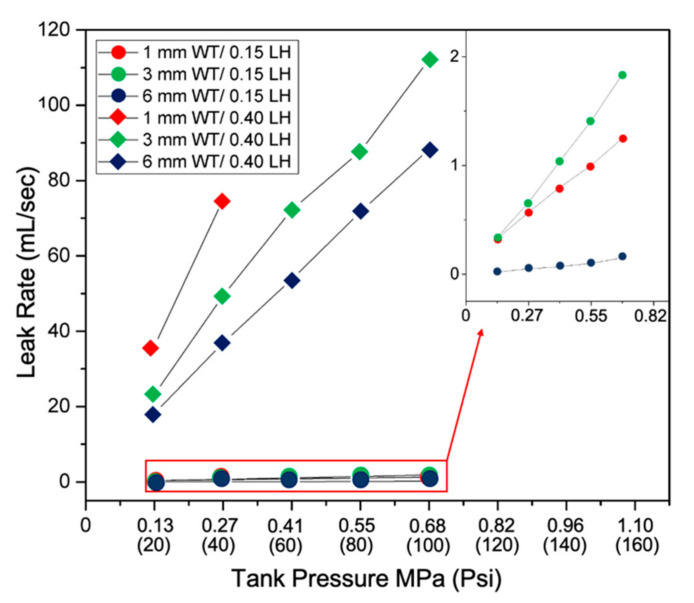
Quantitative vessel volumetric leak rate graph. Top three lines and bottom lines correspond to 0.40 mm 0.15 mm LH respectively.

**Figure 7 materials-13-05328-f007:**
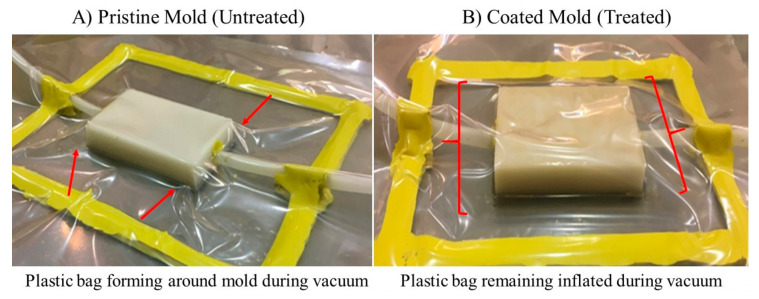
(**A**) pristine and (**B**) coated mold under vacuum to visually evaluate structures porosity.

**Figure 8 materials-13-05328-f008:**
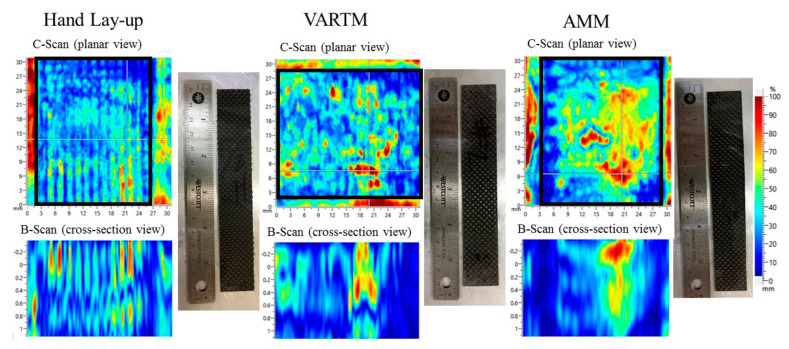
Dolphicam Non-Destructive Evaluation images of HL, VARTM, and AMM composite manufactured samples.

**Figure 9 materials-13-05328-f009:**
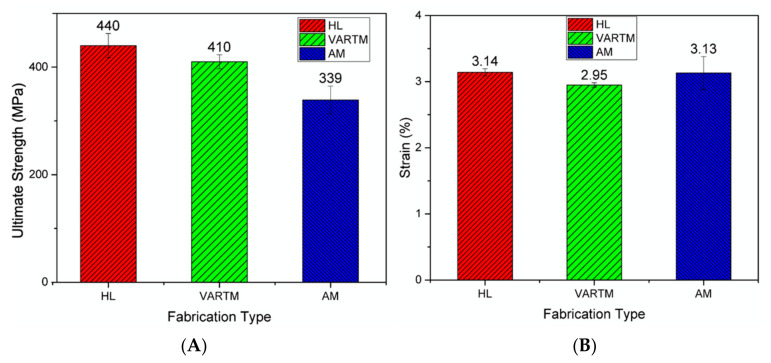
(**A**) Ultimate strength and (**B**) rupture strain data experienced during tensile fracture from HL, VARTM, and AMM composite panels.

**Table 1 materials-13-05328-t001:** Input and Output vessel design parameters.

Input	Layer Height (mm)	0.15			0.40		
	Wall Thickness (mm)	1	3	6	1	3	6
Output	Number of Layers	501	514	534	187	192	200
	Printing Time (minutes)	103	113	152	39	43	57
	Internal volume (cm^3^)	17,809.37					

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
