# Peer review of "Porosity Reducing Processing Stages of Additive Manufactured Molding (AMM) for Closed-Mold Composite Fabrication"

_materials, 2020, doi:10.3390/ma13235328_

Round 1

Reviewer 1 Report

I'm very glad to become acquainted with such interesting research. Hope that my review will help to improve the quality of the article.

Text notes:

Abstract:
Abbreviations should be deleted from the abstract.

2. Materials and Methods:
There are double abbreviations, ex. AMM and HL on lines 46, 79-80, and on lines 154, 309.
The MTS abbreviation is used only once.
SI units should be used.
Paragraph 2.1.1.:
Abbreviations in text and figures should be minimized. It’s better to use abbreviations or full words in the figure. Eg., should abbreviation or full words used in figure 2 caption.
Internal volume in figure 2 and table 1 have different units – mm3 and cm3.

3. Results:
In figure 6 to make it easy to read it’s better to use two types of forms and three types of color. Eg., circles for 0.15 mm LH, diamonds for 0.4 mm LH, and red color for 1 mm WT, green for 3 mm WT, and blue for 6 mm WT. It will be more informative.
Paragraph 3.4.1.:
Composite Processing should be placed into 2. Materials and Methods or even 1. Introduction. Because of my point, 3.4.1 paragraph's main topic is the technical description and differences between HL and RATM methods.
Figure 8 – abbreviation NDE should be deleted from the figure caption.
Bibliographic description for references 1, 5, 6, 13 should be checked.

Research notes:

Paragraph 3.4.3:
It’s not clear what do you mean by using the grip pressure term, because according to ASTM D 3039 tensile tests should be provided.
The term Peak stress is not clear; maybe the ultimate stress term implied?
How many samples were tested for each fabrication type?
What type of failure mode was? Does it change according to different fabrication types?
There is information about failure mode in paragraph 3.4.3 “According to ASTM D 3039, both HL and VARTM samples experienced angeled, splitting, and grip failure modes generally located in the gauge area. AMM samples consistently experienced lateral failures in the gauge area.” But, this description contradicts ASTM D 3039. Thus, “angeled, splitting, and grip failure” are all different types of failure. Does this mean that all samples with the same fabrication types have different types of destruction? Maybe it was so, but it does not clear from the text. Information about the failure area and failure location is missing.
Why the Tensile Modulus of elasticity wasn't calculated? According to the article, all the needed information was get from the tensile test.
The correction of the paragraph will increase interest in the results by specialists in strength and product design.

Author Response

Please see the attachment. Reviewer comments are at the end of document.

Reviewer 2 Report

Interesting paper. Here are my minor comments:

  1. Page 2, line 44 - expand CNC.  Spell out acronyms once in the beginning of the paper  
  2. Conclusions of this manuscript are not adequately supported by experimental evidence.
  3. How did you calculate the fiber volume fraction ratio? 

Round 2

Reviewer 1 Report

I'm very glad to be a reviewer of such a remarkable article. I hope that my review helps you to improve the quality of the article. But, still, there are some shortcomings in the article:

In Figure 6. Psi units should be changed to SI units, I think MPa will be more comfortable for presenting values.

It’s better to write kPa instead of KPa (lines 197, 204, 214, 222, 224, 226, 233)

Line 293: I think, MPa units should be used for ultimate stress values instead of Pa (N/mm2 is equal to MPa).

Line 294: Was it correct to use kPa units for the elasticity modulus values? Maybe GPa should be used?

Figure 9. Peak stress on the axis label of figure 9a should be changed to Ultimate stress, and, I think, that MPa units should be used instead of Pa.

There is a gray outline around Figure 9, I think it's better to delete it.

The mentioned shortcomings do not influence the quality of the article and the results of the research. I can advise the authors to try to use CT for non-destructive evaluation. In my opinion, it's more informative for porosity structure analysis and can expand understanding of the problem.
